# Physical Activity and the Development of Post-Transplant Diabetes Mellitus, and Cardiovascular- and All-Cause Mortality in Renal Transplant Recipients

**DOI:** 10.3390/jcm9020415

**Published:** 2020-02-03

**Authors:** Oyuntugs Byambasukh, Maryse C. J. Osté, António W. Gomes-Neto, Else van den Berg, Gerjan Navis, Stephan J. L. Bakker, Eva Corpeleijn

**Affiliations:** 1Department of Epidemiology, University Medical Center Groningen, University of Groningen, 9713 GZ Groningen, The Netherlands; e.corpeleijn@umcg.nl; 2Department of Internal Medicine, Mongolian National University of Medical Sciences, 976 Ulaanbaatar, Mongolia; 3Department of Nephrology, University Medical Center Groningen, University of Groningen, 9700 RB Groningen, The Netherlands; m.c.j.oste@umcg.nl (M.C.J.O.); a.w.gomes.neto@umcg.nl (A.W.G.-N.); e.van.den.berg@umcg.nl (E.v.d.B.); g.j.navis@umcg.nl (G.N.); s.j.l.bakker@umcg.nl (S.J.L.B.)

**Keywords:** physical activity, renal transplant recipients, transplantation, post-transplant diabetes mellitus, cardiovascular mortality, mortality

## Abstract

(1) Background: Little is currently known about the health impacts of daily-life moderate-to-vigorous physical activity (MVPA) in relation to the development of post-transplant diabetes mellitus (PTDM) and the long-term survival of renal transplant recipients (RTRs). (2) Methods: We analyzed self-reported data on MVPA within non-occupational and occupational domains, estimated with the SQUASH questionnaire, from a prospective cohort study of RTRs (*n* = 650) with a functioning graft exceeding 1 year. PTDM diagnoses were based on plasma glucose levels (≥126 mg/dL), HbA1c (≥6.5%), and the use of antidiabetic medication. Mortality data were retrieved from patient files up to the end of September 2015. (3) Results: During a median follow-up period of 5.3 years, 50 patients (10%) developed PTDM and 129 (19.8%) died. Of these deaths, 53 (8.9%) were caused by cardiovascular disease. Cox regression analyses showed that higher MVPA levels among patients were associated with a lower risk of PTDM (hazard ratio (HR); 95% confidence interval (95%CI) = 0.49; 0.25–0.96, *p* = 0.04), cardiovascular- (0.34; 0.15–0.77, *p* = 0.01), and all-cause mortality (0.37; 0.24–0.58, *p* < 0.001) compared with No-MVPA patients, independently of age, sex, and kidney function parameters. Associations of MVPA with cardiovascular and all-cause mortality remained significant and materially unchanged following further adjustments made for transplant characteristics, lifestyle factors, metabolic parameters, medication use, and creatinine excretion (muscle mass). However, the association between MVPA and PTDM was no longer significant after we adjusted for metabolic confounders and glucose levels. (4) Conclusion: Higher MVPA levels are associated with long-term health outcomes in RTRs.

## 1. Introduction

Renal transplantation is a more effective treatment strategy than chronic dialysis in patients with end-stage renal disease [1]. However, post-transplant patients are at an increased risk of developing cardiometabolic diseases that lead to high morbidity and mortality among renal transplant recipients (RTRs) [2,3]. The cardiovascular mortality rate is estimated to be 10 times higher for RTRs compared with the general population [2]. Moreover, approximately 20% of RTRs develop post-transplant diabetes mellitus (PTDM) [3]. This situation necessitates an investigation aimed at developing strategies for improving the management of long-term health outcomes in RTRs.

Studies have consistently found that physical activity (PA) is a modifiable factor that contributes to reducing the risk of cardiometabolic diseases and premature mortality within the general population [4,5]. However, there is limited data on the impacts of PA on RTRs [6,7,8,9,10]. Studies conducted on the benefits of PA have mostly focused on the intermediate outcomes of clinical trials entailing exercise training programmes [6,7,8]. A few studies found that a low PA is significantly associated with substantial weight gain and with risks of cardiovascular and all-cause mortality in RTRs [11,12,13,14]. Moreover, it remains unclear whether the benefits of increased PA in relation to patients’ long-term outcomes are independent of their health and transplant characteristics (i.e., kidney function and duration of pre-transplant dialysis), lifestyle factors, and use of medication [11,12,13].

Clinical guidelines for the general population recommend the performance of at least 150 min of moderate-to-vigorous physical activity (MVPA) per week [15]. There are no specific clinical guidelines or recommendations for RTRs. A recent position statement on exercise for solid organ transplant recipients released in 2019 recommending that it is a key step toward raising awareness of the importance of exercise training in the patients among transplant professionals [16]. The available data show that the level of daily-life PA is lower for RTRs compared with individuals within the general population [17,18,19]. It is unclear whether individuals within the general population as well as RTRs can attain the recommended MVPA level by engaging in different domains of daily-life activities, such as non-occupational and occupational activities. Results showed that occupational MVPA should not be included within assessments of healthy daily-life PA and should not be deemed a substitute for leisure time MVPA. Specifically, it is not known whether both non-occupational and occupational PA can contribute to the improved health of RTRs [20,21,22].

Therefore, we aimed to investigate the association between daily-life MVPA and the risk of developing long-term health outcomes, such as PTDM as well as cardiovascular and all-cause mortality in RTRs sampled from a large prospective study. We also examined whether these associations were independent of several variables, including age, sex, kidney function, transplant characteristics, lifestyle factors, medication use, metabolic parameters, and anthropometric measures. Moreover, we assessed the benefits of non-occupational MVPA as well as total daily-life MVPA, including occupational PA.

## 2. Methods

### 2.1. Study Population

This study was conducted in a large single-center prospective cohort of stable outpatient RTR [23,24]. A total of 817 adult RTRs who met the study’s eligibility criteria, namely having a functioning graft for at least 1 year and no history of alcohol and/or drug addiction, were invited to participate in the study. Further exclusions were apparent systemic diseases, such as malignancies or active infections. In total, 707 (86.5%) RTRs signed written informed consent. Baseline data were collected between November 2008 and May 2011. We excluded 57 RTRs, whose PA questionnaires were incomplete, from the analysis, leaving a total of 650 RTRs. Subsequently, 148 and 61 RTRs, respectively, with a history of diabetes and cardiovascular diseases (CVDs) prior to undergoing transplants, were excluded from the analyses of PA and the development of PTDM and cardiovascular mortality. The study was conducted according to the Helsinki Declaration and was approved by the UMCG’s review board (METc 2008/186).

### 2.2. Measurements at Baseline

All baseline measurements have been previously described in greater detail elsewhere [25]. Body mass index (BMI) was calculated as weight (kg) divided by height squared (m^2^). A semiautomatic device (Dinamap^®^1846; Critikon, Tampa, FL, USA) was used to measure blood pressure in a half-sitting position and the average of the final three readings of blood pressure was used. Information on medication was derived from patient records. Daily caloric intake and alcohol consumption were calculated from a validated Food Frequency questionnaire. Information on smoking was obtained by a questionnaire. The serum creatinine-based Chronic Kidney Disease Epidemiology Collaboration equation was used to calculate the estimated glomerular filtration rate (eGFR) [26]. Creatinine excretion—a marker of muscle mass—was calculated from the 24-h urine collection as described earlier [27].

### 2.3. Assessment of Physical Activity

The SQUASH is a validated questionnaire used to estimate habitual physical activities performed during a normal week [28]. The SQUASH is pre-structured into four domains: commuting, leisure time and sports, household, and occupational activities. Questions consisted of three main queries: days per week, average time per day, and intensity. In this study, we used activities at the moderate (4.0–6.5 MET) to vigorous (≥6.5 MET) level. Metabolic equivalent (MET) values were assigned to activities according to Ainsworth’s Compendium of Physical Activities [29]. We used the combination of leisure-time and commuting (non-occupational) moderate-to-vigorous physical activity (MVPA) minutes per week (min/week) as a measure of PA in this study, since active commuting of high intensity and longer duration is often replacing sports activities, like cycling. We did not include occupational MVPA in the main analysis because of its health benefit is not clear in the general population [20,21,22,30]. In an additional analysis, we investigated the association between total MVPA, including occupational MVPA, with clinical endpoints. Participants were subdivided into three categories based on their levels of non-occupational MVPA. RTRs who did not engage in PA at a moderate-to-vigorous level were deemed ‘inactive’ (no-MVPA), and the remaining participants (MVPA > 0) were divided into two groups based on median values of non-occupational MVPA (less active, MVPA-1 and active, MVPA-2). The MVPA min/week (median, interquartile range (IQR)) was used to define the MVPA groups (MVPA-1 and MVPA-2): 5-197 (120, 60–150) and 200–1680 (360, 260–540).

### 2.4. Endpoints of the Study

Endpoints of interest in this study were post-transplant diabetes mellitus (PTDM), cardiovascular and all-cause mortality. PTDM was defined according to the presence of at least one of the following criteria: diabetes symptoms (e.g., polyuria, polydipsia, or unexplained weight loss) along with a non-fasting plasma glucose concentration of ≥200 mg/dL (11.1 mmol/L); fasting plasma glucose concentration (FPG) ≥ 126 mg/dL (7.0 mmol/L); start of antidiabetic medication; or HbA1c ≥ 6.5% (48 mmol/L). This definition matched the diagnostic criteria for diabetes applied by the American Diabetes Association, including HbA1c levels, as proposed by the expert panel constituted at the international consensus meeting on PTDM [31,32]. Cardiovascular mortality and all-cause mortality were monitored through continuous surveillance conducted within the outpatient program and retrieved from patients’ files up to the end of September 2015. No participants were lost to follow-up.

### 2.5. Statistical Analysis

The patient characteristics are expressed as means with a standard deviation for normally distributed variables or as medians with interquartile range (25th to 75th percentile) for non-normally distributed variables and numbers with percentages in case of categorical data. The differences between groups were tested using 1-way analysis of variance tests or Kruskal–Wallis tests for normally and non-normally distributed continuous variables, respectively. The frequency distributions of categorical variables were analyzed using the Pearson Chi-Square test.

We adopted MVPA as a continuous and categorical variable in a Cox regression analysis. First, we tested associations of non-occupational MVPA, considered as a continuous variable, on clinical endpoints. In this analysis, MVPA (measured in min/week) was log-transformed to obtain a normal distribution. Thereafter, multivariate Cox regression analyses were performed to examine whether higher non-occupational MVPA is associated with lower risks of PTDM, cardiovascular mortality, and all-cause mortality independently of potential confounders which are clinically known confounders in the relating associations of physical activity with long-term outcomes [6,7,12,13,14,33]. In these analyses, we first adjusted for age and sex (model 1) as well as kidney function parameters, including eGFR, proteinuria, the time lapse between transplantation and the baseline measures, and primary renal disease (model 2). We further adjusted model 1 for transplant characteristics (acute rejection, preemptive transplantation, and living donor status) in model 3. Similarly, we adjusted model 1 for lifestyle factors, such as smoking, alcohol consumption, and daily caloric intake (model 4); calcineurin inhibitors and prednisolone used as immunosuppressive medication (model 5); systolic blood pressure, use of antihypertensive drugs, high-density lipoprotein cholesterol (HDL), and triglycerides (model 6), BMI and waist circumference (model 7), and 24-h creatinine excretion (model 8). With regard to potential collinearity (model 7), we tested the correlation between BMI and waist circumference (*r* = 0.84, *p* < 0.001 for men and *r* = 0.81, *p* < 0.001 for women). Then we performed Cox-regression analyses using separate models adjusted for BMI and waist circumference separately (Appendix A. We found no differences relating to the confounding effects of total fat and fat distribution. Finally, we included those variables in the same model. In addition, we adjusted model 1 for baseline hemoglobin A1C and fasting plasma glucose (model 9) relating to the association between MVPA and PTDM. We also adjusted model 1 relating to the association between MVPA and PTDM for diet quality (Model 10). We furthermore investigated whether diet quality might modify the association of MVPA with development of PTDM by additional inclusion of a product-term of the continuous variables of diet quality and MVPA in the concerned model, to assess potential interaction between the two. Mediterranean diet score was used as diet quality and assessed with a 177-item validated food frequency questionnaire which is described in greater detail elsewhere [33]. All models (1-10) include up to 6 variables to fulfil the rule of thumb which allows 1 variable per 7–10 events. This is now fulfilled for all analyses [34,35]. Hazard ratios were reported with 95% confidence intervals. Proportional hazard assumptions were tested using the Schoenfield residuals method developed by Grambsch and Therneau [36]. Penalized splines were constructed to visualize the association of non-occupational MVPA with PTDM as well as cardiovascular and all-cause mortality independently of age and sex.

We performed additional analyses to explore the role of work within this population by investigating the associations between total MVPA, including occupational MVPA, and non-occupational MVPA with clinical endpoints for the RTRs who worked (*n* = 322, 49.5%). Occupational status was defined using the answers for the questions related to occupational PA. If responders answered as not applicable, we considered them as unemployed. Another subgroup analysis was performed to address changes in these associations across age categories. The population was categorized as being over or under 55 years of age, based on the WHO guideline on the prevention of CVD [37]. Finally, to rule out competing mortality risks associated with the occurrence of PTDM, we conducted competing risk analyses following the procedures outlined by Fine and Gray [38].

A two-sided statistical significance was set at *p* < 0.05 for all tests. All statistical analyses were performed using SPSS software V.22 (IBM Inc., Chicago, IL, USA,) R software V.3.2.2 (R Foundation for Statistical Computing, Vienna, Austria), STATA version 13.0 (StataCorp LP, College Station, TX, USA) and Graph Pad Prism 7 (Graph Pad Software Inc., La Jolla, CA, USA).

## 3. Results

### 3.1. Baseline Characteristics

A total of 650 RTRs (men: 56.3%, mean age: 51.8 ± 13.2 years old) were examined in this study. Baseline measurements were taken 5.7 years (median value; interquartile range (IQR): 1.9–12.1 years) post-transplantation. Of the total sample of RTRs, 37.8% (*n* = 246) did not perform daily MPVA at all within any domain. The other RTRs spent a median of 200 min (IQR = 120–360 min per week) engaged in non-occupational MVPA. Table 1 shows the baseline characteristics of RTRs according to their non-occupational MVPA levels. RTRs in the active groups (MVPA > 0) had lower values for BMI, waist circumference, and systolic blood pressure and higher creatinine excretion values compared with the values of the inactive group (no-MVPA). Moreover, higher alcohol consumption, lower concentrations of triglycerides and HDL-C, haemoglobin A1C, and less proteinuria and diabetes at the baseline level along with more ‘living donors’ were observed for the ‘active’ groups compared with the ‘inactive’ group. Appendix A presents the baseline characteristics of the RTRs according to the presence of clinical endpoints. Appendix A further shows levels of daily MVPA according to the participants’ ages and work status. As expected, total MVPA values, including those for occupational MVPA, were significantly higher in RTRs who were working (*n* = 322). However, when working status or age was considered, the levels of non-occupational MVPA did not differ significantly.

### 3.2. Post-Transplant Diabetes Mellitus

A total of 50 RTRs (10%) had developed PTDM after a median follow-up period of 5.3 years (4.1–6.0 years). The multivariable Cox proportional hazard models showed that the group with the highest level of non-occupational MVPA was associated with a lower risk of PTDM (hazard ratio (HR); 95% CI = 0.49; 0.25–0.96, *p* = 0.04) compared with the no-MVPA group, independently of age, sex, and kidney function parameters (model 1, Table 2). This association remained significant after we made further adjustments for kidney function parameters, transplant characteristics, lifestyle factors, Mediterranean diet score and 24-h creatinine excretion quantities (considered as a marker of muscle mass) (models 2–4, 8, and 10). Following adjustments made for immunosuppressive medication (model 5), metabolic parameters (model 6), anthropometric measures (model 7), and baseline glucose levels (model 9), the highest level of MVPA was no longer associated with PTDM. However, when MVPA was applied as a continuous variable in the Cox regression analysis, as opposed to using groups of MVPA levels, a higher non-occupational MVPA was associated with a lower risk of PTDM independent of all of the above-mentioned confounders apart from the adjustment of metabolic parameters and glucose level.

### 3.3. Cardiovascular and All-Cause Mortality

During the follow-up period, 129 (19.8%) patients died. Of these deaths, 53 (8.9%) were caused by cardiovascular disease (CVD). In the multivariable Cox proportional hazard models, the highest level of non-occupational MVPA was associated with a lower risk of cardiovascular mortality (HR; 95% CI = 0.34; 0.15–0.77, *p* = 0.01) compared with the no-MVPA group, independently of age, sex, and kidney function parameters (model 1–2, Table 2). This association remained significant after further adjustments were made for transplant characteristics, immunosuppressive medication, metabolic parameters, and anthropometric measures (models 3, 5–7). However, the association was no longer significant after adjusting for lifestyle factors (model 4) and creatinine excretion (model 8). Moreover, the association of MVPA with cardiovascular mortality was sustained independently of all of the potential confounders when non-occupational MVPA was applied as a continuous variable in the Cox regression (models 1–9, Table 2).

With regard to all-cause mortality, the group with the highest level of non-occupational MVPA was associated with a lower risk of all-cause mortality (HR; 95% CI = 0.37; 0.24–0.58, *p* < 0.001) compared with the no-MVPA group (model 1, Table 2). This association remained significant after we adjusted for potential confounders (models 2–8). However, the association weakened after we adjusted for transplant characteristics (model 3), metabolic parameters (model 6), and creatinine excretion (model 8). When log-transformed non-occupational MVPA was applied as a continuous variable in the Cox regression analysis, the association was independent of all of the above-mentioned confounders, apart from creatinine excretion (model 8), remaining materially unchanged.

To illustrate these associations further, age-and sex-adjusted penalized splines and the Kaplan-Meier survival curves are shown in Figure 1 and Figure 2.

### 3.4. Additional Analyses

Additional analyses of the subgroup of working RTRs revealed that the inclusion of occupational PA in the estimate of MVPA resulted in the attenuation of the HRs of all of the significant and non-significant associations (Table 3). The association of non-occupational MVPA with cardiovascular and all-cause mortality was stronger compared with that of total MVPA, which includes occupational MVPA.

Age-stratified analyses revealed that associations of MVPA with long-term health outcomes were stronger in older adults (Figure 3).

The results of the competing risk analyses showed that there was no strong influence of a competing risk of all-cause mortality on the association of MVPA with PTDM. For instance, the competing HR was 0.51 (0.30–0.92, *p* = 0.04) for the highest MVPA with PTDM after adjusting for age and sex. By comparison, the HR was 0.49 (0.25–0.96, *p* = 0.04, model 1, Table 2) when competing risks were discounted.

## 4. Discussion

We found that increased daily-life MVPA is associated with a reduced risk of PTDM, cardiovascular mortality, and all-cause mortality in RTRs independently of age, sex, baseline kidney function parameters, transplant characteristics, and other lifestyle habits. The association of MVPA with PTDM was affected by the adjustments we made for baseline glucose levels and metabolic parameters, but it did not seem to be affected by other potential confounders, notably anthropometric and immunosuppressive medication. The associations of MVPA with cardiovascular and all-cause mortality were not substantially affected by adjustments made for the above-mentioned confounders. These results confirm the importance of PA in the long-term healthcare management of RTRs.

Previous studies have found that PTDM is highly prevalent in RTRs [3,39]. However, data on lifestyle interventions for improving glucose tolerance or observational data on the association of increased PA with incidences of PTDM are lacking [13,40]. An intervention study showed that lifestyle modifications, including the incorporation of exercise training, improved 2-h postprandial glucose levels in RTRs who were glucose intolerant [40]. One observational study found that higher levels of PA are associated with a lower risk of glucose intolerance in RTRs [13]. However, this study entailed a cross-sectional design and did not test whether this association of PA is independent of other potential confounders. In our longitudinal study, the association of MVPA with PTDM was found to be independent of age, sex, baseline kidney function parameters, transplant characteristics, and other lifestyle factors, such as smoking, alcohol use, and diet (daily caloric intake and Mediterranean diet score). However, the association was affected by adjustments made for immunosuppressive medication, anthropometric measures (BMI and waist circumference), baseline glucose levels, and metabolic parameters. It is widely accepted that obesity is associated with the development of diabetes within the general population [41]. The use of immunosuppressive medications play a role in the development of PTDM through a pathway of stimulation of gluconeogenesis affecting increased blood glucose which can leads to insulin resistance in combination with other mechanisms [39]. However, when we applied log-transformed continuous MVPA, significant associations were observed after we adjusted for immunosuppressive medication and anthropometric measures, indicating that statistical power issues may also play a role. Thus, further large-scale studies of a longer duration should be conducted to explore whether or not MVPA is associated with PTDM independently of immunosuppressive medication and obesity. Furthermore, diet is an important factor in the development of diabetes. A previous analysis by Osté et al. for our study population showed that Mediterranean style diet predicts the development of PTDM [33]. We found that the association between MVPA and PTDM became slightly stronger when adjusted for Mediterranean diet score indicating the importance of diet, but there was no effect modification by diet quality (P-interaction = 0.147).

A previous study, investigating another sample of RTRs, found that a lower PA is strongly associated with an increased risk of cardiovascular and all-cause mortality [14]. In their Cox regression analyses, these authors found that the association was independent of potential confounders, including the history of CVD, muscle mass, and Framingham CVD risk score factors. However, they did not adjust for some clinical variables, such as kidney function and transplant characteristics (e.g., transplant vintage and donor type). Our study supports an independent association of PA with the risk of cardiovascular and all-cause mortality. Many studies have pointed to the benefit of PA within the general population in preventing premature mortality [4,5]. One of the mechanisms proposed to explain the effects of increased PA entails the improvement of all organ systems, especially the cardiovascular system. Specifically in RTRs, improved cardiovascular function is associated with improvements in kidney function. Increased physical activity can support perfusion and oxygen delivery in the kidneys. Studies have shown that higher levels of daily-life PA are associated with a lower risk of renal function decline within the general population and in patients with chronic kidney disease [42,43,44]. Consequently, increased PA, by improving kidney function, may be of benefit for long-term graft survival. This effect may also be due to improvements in metabolic dysfunctions, such as insulin resistance, impaired glucose tolerance, dyslipidemia, and hypertension, all of which are related to (central) adiposity [45,46,47]. Furthermore, a number of studies on diet analysed in-depth the effect of dietary factors on the same outcomes such as PTDM, renal function decline and mortality [23,24,25,33,48]. They suggest that lifestyle is very important for RTR, however, it should be noted that MVPA in daily life has not gotten that much attention. Taken together, these findings suggest that the improvement of daily-life MVPA needs to be evaluated as a therapy for improving patients’ long-term survival.

Within this RTR population, MVPA levels were lower than those within the general population. In our study, 38% of RTRs were inactive (no-MVPA), whereas in the Lifelines cohort, a population-based study for which the same questionnaire (SQUASH) and comparable data processing methods were used, the prevalence of inactivity (no-MVPA) was 10% (*n* = 125,402, 40.5% males, median age of 45) [30,49]. Even in different age groups and gender, it was lower, ranging between 7.5% (*n* = 42,661, 40% of males, median age of 40) and 12.5% (*n* = 34,506, 45.6% of males, median age of 56) in the Lifelines. Lower PA levels among RTRs may be attributed to lower muscle mass (a structural abnormality) and muscle weakness (a functional abnormality) [6]. Our descriptive analysis indicated that inactive RTRs had a lower 24-h creatinine excretion value (a marker of muscle mass) compared with that of active RTRs. We also found that the duration of pre-transplantation dialysis was longer in inactive RTRs, although not significantly so. Studies concluded that low muscle mass can be caused by low PA levels [6,7,9]. This conclusion is in line with our findings, indicating that the association was slightly attenuated after we adjusted for renal factors and muscle mass but that the effect of PA remained evident. Actually a shorter time on dialysis would thus also help post-transplant health because studies showed that the level of PA declines in patients with end-stage kidney diseases and it increases after transplantation [7]. Finally, both recovery of activity after transplantation, as well as prevention of inactivity and loss of muscle mass in people with longstanding kidney disease is important for long-term health after transplantation.

A growing body of evidence is showing that occupational MVPA may have no clear benefit on health in the general population [20,21,22,30,49]. This was tested in our study including a specific patient population, the RTR. Even in the case of RTR, where being at work may be indicative of relatively good health, individuals who were much more active in terms of their occupational MVPA may not obtain any additional benefits for health. A clear mechanism that prevents occupational PA from generating health benefits is missing. There is always the possibility of residual confounding by factors such as sex, socioeconomic status, work-related stress, and body weight in the association between occupational PA and health outcomes [20,22,30,49]. Studies attempted to explore the possibility of residual confounding, but also found no clear association of occupational MVPA and health outcomes. Thus, we suggest that it is important to be aware that occupational MVPA should not be considered as a substitute for leisure time MVPA in RTR.

The potential benefit of PA seems to be more pronounced in older adults, a phenomenon that was described before in the general population [50]. In the general population, studies concluded that the benefit of PA can be gained more easily when there is more room for improvement, like as in older people. However, it might also be that its effects will be potentially outweighed by other, more important clinical factors (e.g., comorbidities and medication use). Therefore, we attempted to test the effect of physical activity in specific groups such as in RTR in two age groups. We found that a higher MVPA is strongly associated with the development of long-term outcomes such as PTDM and cardiovascular mortality in younger and older adults, but is especially stronger in older adults. Thus, older RTR who are able to remain active despite their longstanding condition are likely to remain relatively healthy.

The strengths of this study include its prospective design, long duration, and complete follow-up. Another strength is we included stable RTRs after transplantation and studied relevant clinical outcomes. Nevertheless, there are some limitations to our study. The observational nature of the study precludes us from drawing conclusions regarding causality. A limitation of this study was its use of self-reporting, which is subject to recall bias, for the PA assessment. However, the SQUASH questionnaire has been validated within general as well as specific populations, such as patients who have undergone total hip arthroplasty and those with ankylosing spondylitis [28,51,52]. Furthermore, PA was assessed at a single point in time. However, in RTR, after 1-year of transplantation, PA is increased by 30% and remained materially unchanged the next 5-years [18]. In this study, we included RTR > 1 year graft functioning with a median of 5.7 years post-transplantation. Another limitation is that we could not fully control for the history of all cardiometabolic diseases in the association of MVPA with all-cause mortality. Patients with a history of diabetes or CVD before the transplantation were excluded from the analyses on the association between MVPA and PTDM or CV mortality. However, cardiometabolic diseases might be more prevalent in ‘No-MVPA’ group after transplantation as well. A limitation is that we could not have data on functional evaluations, like e.g., a 6 min walking test, which could have provided important information on cardiovascular efficiency. Finally, single-center nature of study, which mainly consisted of white people is unclear whether our findings can be extrapolated to other populations. It would be relevant to repeat our study in other patient populations.

## 5. Conclusions

Higher daily-life MVPA is associated with a reduced risk of PTDM as well as cardiovascular and all-cause mortality in RTRs, suggesting that PA has a positive influence on the long-term health management of RTRs. The associations of MVPA with cardiovascular and all-cause mortality were not substantially affected by adjustments made for potential confounders, such as age, sex, baseline kidney function parameters, transplant characteristics, lifestyle habits, metabolic parameters, anthropometric measures, and immunosuppressive medication. The association of MVPA with PTDM was affected by adjustments of metabolic parameters and glucose levels. The potentially beneficial effects of daily-life PA apply to non-occupational activities at the moderate-to-vigorous level (e.g., commuting, leisure activities, or sport). By contrast, a higher level of occupational MVPA is not directly associated with the development of long-term outcomes. The associations of non-occupational MVPA and the risk of PTDM and cardiovascular mortality were also stronger in older adults. Finally, we suggest that because of the long-term importance of PA, it should be embedded in the healthcare management of RTRs. Furthermore, large scale interventional studies are needed to test the ab initio effect of physical activity after transplantation on the development of post-transplant diabetes mellitus.

## Figures and Tables

**Figure 1 jcm-09-00415-f001:**
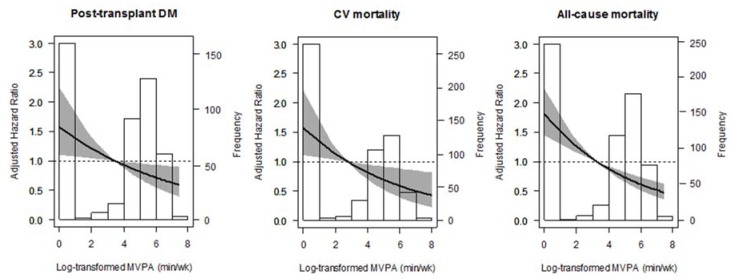
Association between non-occupational MVPA and post-transplant diabetes mellitus (PTDM), Cardiovascular (CV)mortality, and all-cause mortality in renal transplant recipients (RTRs).

**Figure 2 jcm-09-00415-f002:**
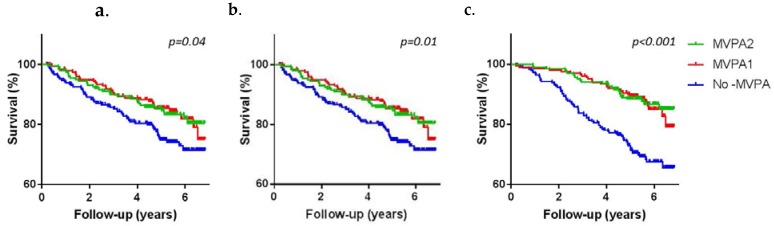
Probability of survival for PTDM (**a**), cardiovascular mortality (**b**), and all-cause mortality (**c**) according to non-occupational MVPA level.

**Figure 3 jcm-09-00415-f003:**
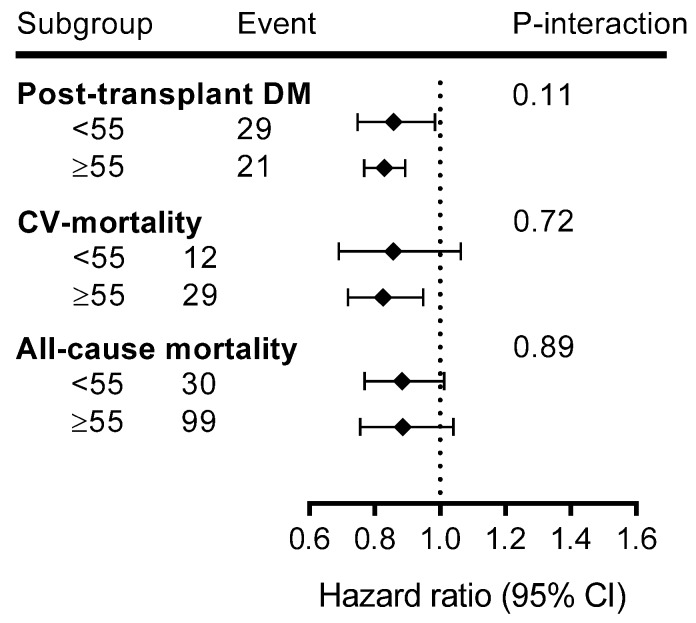
Subgroup analysis for the associations of MVPA with long-term health outcomes over age categories.

**Table 1 jcm-09-00415-t001:** Characteristics of the study population, according to MVPA level.

Variable	Total (*n* = 650)	No-MVPA (*n* = 246)	MVPA-1 (*n* = 201)	MVPA-2 (*n* = 203)	*p*-Value
Age (years)	52.6 ± 12.8	54.1 ± 11.8	51.9 ± 13.4	51.6 ± 13.3	0.08
Male gender (%, n)	56.3 (366)	56.1 (138)	55.7 (112)	57.1 (116)	0.96
Current smoking (%, n)	12.3 (80)	15.3 (36)	12.4 (25)	9.5 (19)	0.20
Occupational status: Employed * (%, n)	49.5 (322)	44.7 (110)	51.2 (103)	53.7 (109)	0.14
Alcohol use (g/day)	2.61 (0.1–11.1)	1.45 (0.1–9.7)	2.51 (0.1–9.7)	3.95 (0.1–14.1)	0.01
Total energy intake (kcal/d)	2174.9 ± 640.7	2114.7± 720.4	2247.5 ± 598.7	2173.2 ± 573.9	0.11
Non-occupational MVPA (min/week)	90 (0–240)	0	120 (60–150)	360 (260–540)	−
Anthropometric measures					
Body mass index (kg/m^2^)	26.7 ± 4.84	27.9 ± 5.49	25.7 ± 4.26	26.1 ± 4.23	0.001
Waist circumference, men (cm)	101.1 ± 13.4	104.0 ± 13.7	98.9 ± 13.5	99.7 ± 12.4	0.01
Waist circumference, women (cm)	95.0 ± 15.8	99.7 ± 12.4	91.5 ± 14.6	93.1 ± 14.1	0.01
Creatinine excretion (mmol/24h)	11.7 ± 3.49	11.2 ± 3.80	11.8 ± 3.29	12.2 ± 3.23	0.01
Lipids and blood pressure					
Total cholesterol (mmol/L)	5.14 ± 1.11	5.18 ± 1.18	5.17 ± 1.10	5.07 ± 1.02	0.48
Triglyceride (mmol/L)	1.68 (1.2–2.33)	1.85 (1.3–2.6)	1.67 (1.2–2.4)	1.59 (1.2–2.05)	0.001
HDL-C in men (mmol/L)	1.27 ± 0.41	1.23 ± 0.41	1.27 ± 0.36	1.32 ± 0.45	0.21
HDL-C in women (mmol/L)	1.56 ± 0.51	1.39 ± 0.43	1.62 ± 0.54	1.70 ± 0.53	0.001
Systolic blood pressure (mm Hg)	136.2 ± 17.3	138.3 ± 18.5	135.6 ± 16.7	134.2 ± 16.3	0.04
Diastolic blood pressure (mm Hg)	82.8 ± 10.9	83.1 ± 11.2	82.5 ± 11.3	82.6 ± 10.3	0.77
Cardiovascular medication use					
Antihypertensive (%, n)	88 (572)	92.7 (228)	81.1 (163)	89.2 (181)	0.001
A2 antagonist (%, n)	14.8 (96)	15.4 (38)	12.9 (26)	15.8 (32)	0.68
ACE inhibitor (%, n)	32.2 (209)	32.1 (79)	30.3 (61)	34.0 (69)	0.74
RAAS blockers (%, n)	47.8 (311)	49.2 (121)	44.8 (90)	49.3 (100)	0.58
Beta-blockers (%, n)	63.2 (411)	63.4 (156)	62.2 (125)	64.0 (130)	0.93
Calcium channel blockers (%, n)	24.5 (159)	26.0 (64)	20.4 (41)	26.6 (54)	0.27
Diuretics (%, n)	40.0 (260)	52.0 (128)	27.4 (50)	37.9 (77)	0.001
Vitamin K antagonist (%, n)	11.4 (74)	13 (32)	10.4 (21)	10.3 (21)	0.60
mTOR inhibitor (%, n)	1.8 (12)	3.3 (8)	1 (2)	1 (2)	0.60
Anti-diabetic drugs (%, n)	14.8 (96)	18.7 (46)	14.4 (29)	10.3 (21)	0.045
Statin (%, n)	51.8 (337)	54.9 (135)	52.7 (106)	47.3 (96)	0.27
Glucose metabolism					
Fasting plasma glucose (mmol/L)	5.67 ± 1.82	5.78 ± 192	5.76 ± 2.13	5.46 ± 1.28	0.13
Heamoglobin A1C (%)	5.94 ± 0.78	6.03 ± 0.77	5.94 ± 0.90	5.83 ± 0.65	0.021
Kidney function					
eGFR (mL/min/1.73m^2^)	52.0 ± 20.2	49.9 ± 22.1	53.8 ± 18.7	52.9 ± 18.8	0.09
Albumin excretion (mg/24h)	267.3 ± 734.6	307.2 ± 777.5	175.1 ± 378.5	308.7 ± 917.5	0.11
Proteinuria (%, n)	21.5 (140)	28.0 (69)	16.9 (34)	18.2 (37)	0.01
Primary renal disease (%, n)					0.01
Glomerulosclerosis	28.8 (187)	30.1 (74)	28.4 (57)	27.6 (56)	
Glomerulonephritis	7.7 (50)	5.7 (14)	8.0 (16)	9.9 (20)	
Tubulointerstitial nephritis	11.8 (77)	9.8 (24)	12.9 (26)	13.3 (27)	
Polycystic kidney disease	20.9 (136)	20.7 (51)	19.9 (40)	22.2 (45)	
Renal hypodysplasia	3.5 (23)	4.1 (10)	3.0 (6)	3.4 (7)	
Renavascular diseases	5.7 (37)	7.7 (19)	4.0 (8)	4.9 (10)	
Diabetes mellitus	4.6 (30)	6.5 (16)	5.5 (11)	1.5 (3)	
Others	16.9 (110)	15.4 (38)	18.4 (37)	17.2 (35)	
Duration of dialysis before the transplantation (months)	25 (8–48)	29 (11–51)	19 (4–49)	25 (9–43)	0.51
Transplant characteristics					
Transplant vintage (months)	14.0 (2.0–39.5)	17.0 (2.0–41.0)	12.0 (2.0–44.8)	16.0 (0.5–41.0)	0.49
Cold ischemia time (h)	15.2 (2.8–21.1)	16.4 (3.6–22.0)	15.1 (2.6–21.3)	13.6 (2.5–20.5)	0.10
Living donor (%, n)	34.8 (226)	26.4 (65)	37.3 (75)	42.4 (86)	0.001
Pre-emptive transplant (%, n)	16.6 (108)	13.4 (33)	20.9 (42)	16.3 (33)	0.11
Acute rejection	27.2 (177)	27.2 (67)	27.9 (56)	26.6 (54)	0.96
Immunosuppressive medication					
Calcineurin inhibitor (%, n)	58.3 (379)	59.8 (147)	59.2 (119)	55.7 (113)	0.52
Proliferation inhibitor (%, n)	82.6 (537)	80.1 (197)	84.1 (169)	84.2 (171)	0.62
Prednisolone dose (mg)	10.0 (7.5–10.0)	10.0 (7.5–10.0)	10.0 (7.5–10.0)	10.0 (7.5–10.0)	0.48

Data are presented as mean ± SD or median (interquartile range) and percentage (%, number). MVPA = moderate-to-vigorous physical activity, HDL-C = high-density lipoprotein cholesterol, eGFR = estimated glomerular filtration rate, A2 = angiotensin 2, ACE = angiotensin-converting-enzyme, RAAS = renin–angiotensin–aldosterone system, mTOR = mammalian target of rapamycin. * the number of patients that do have employment.

**Table 2 jcm-09-00415-t002:** Association of non-occupational MVPA with long-term health outcomes.

Physical Activity	MVPA (cont.)		No-MVPA (Ref)	MVPA-1		MVPA-2	
	HR (95% CI)	*p*		HR (95% CI)	*p*	HR (95% CI)	*p*
Post-transplant DM							
No. of events	50/502		23	14		13	
Model 1	0.88 (0.79–0.97)	0.01	1.00	0.57 (0.29–1.10)	0.09	0.49 (0.25–0.96)	0.04
Model 2	0.88 (0.79–0.98)	0.02	1.00	0.61 (0.31–1.20)	0.15	0.49 (0.25–0.96)	0.04
Model 3	0.88 (0.79–0.98)	0.02	1.00	0.55 (0.28–1.07)	0.08	0.48 (0.24–0.95)	0.04
Model 4	0.87 (0.79–0.97)	0.01	1.00	0.57 (0.29–1.12)	0.10	0.46 (0.22–0.94)	0.03
Model 5	0.89 (0.80–0.99)	0.03	1.00	0.59 (0.30–1.26)	0.11	0.52 (0.26–1.03)	0.06
Model 6	0.91 (0.82–1.01)	0.09	1.00	0.70 (0.36–1.40)	0.31	0.60 (0.29–1.22)	0.16
Model 7	0.88 (0.79–0.99)	0.03	1.00	0.63 (0.31–1.25)	0.19	0.50 (0.25–1.03)	0.06
Model 8	0.87 (0.79–0.97)	0.01	1.00	0.55 (0.29–1.08)	0.08	0.47 (0.24–0.93)	0.03
Model 9	0.91 (0.82–1.01)	0.12	1.00	0.72 (0.36–1.41)	0.34	0.59 (0.30–1.19)	0.14
Model 10	0.87 (0.78–0.97)	0.01	1.00	0.58 (0.32–1.25)	0.11	0.44 (0.21–0.92)	0.03
Cardiovascular mortality							
No. of events	53/589		26	14		13	
Model 1	0.84 (0.74–0.94)	0.01	1.00	0.45 (0.22–0.94)	0.03	0.34 (0.15–0.77)	0.01
Model 2	0.84 (0.75–0.95)	0.01	1.00	0.49 (0.23–1.02)	0.06	0.35 (0.16–0.80)	0.01
Model 3	0.86 (0.76–0.96)	0.01	1.00	0.51 (0.25–1.05)	0.07	0.40 (0.18–0.91)	0.03
Model 4	0.87 (0.77–0.98)	0.02	1.00	0.56 (0.26–1.21)	0.14	0.43 (0.19–0.94)	0.046
Model 5	0.84 (0.74–0.94)	0.001	1.00	0.45 (0.21–0.93)	0.03	0.36 (0.16–0.81)	0.01
Model 6	0.85 (0.76–0.96)	0.001	1.00	0.49 (0.23–1.02)	0.06	0.38 (0.17–0.86)	0.02
Model 7	0.85 (0.75–0.96)	0.01	1.00	0.51 (0.23–1.11)	0.09	0.40 (0.17–0.92)	0.03
Model 8	0.87 (0.77–0.98)	0.02	1.00	0.55 (0.26–1.16)	0.12	0.44 (0.19–0.99)	0.051
All-cause mortality							
No. of events	129/650		76	27		26	
Model 1	0.84 (0.78–0.89)	<0.001	1.00	0.39 (0.25–0.61)	<0.001	0.37 (0.24–0.58)	<0.001
Model 2	0.85 (0.79–0.91)	<0.001	1.00	0.43 (0.27–0.67)	<0.001	0.40 (0.26–0.63)	<0.001
Model 3	0.85 (0.79–0.91)	<0.001	1.00	0.41 (0.27–0.64)	<0.001	0.41 (0.26–0.64)	<0.001
Model 4	0.83 (0.77–0.89)	<0.001	1.00	0.41 (0.21–0.64)	<0.001	0.35 (0.22–0.58)	<0.001
Model 5	0.83 (0.78–0.89)	<0.001	1.00	0.39 (0.25–0.61)	<0.001	0.37 (0.23–0.58)	<0.001
Model 6	0.85 (0.79–0.91)	<0.001	1.00	0.42 (0.27–0.66)	<0.001	0.41 (0.26–0.65)	<0.001
Model 7	0.84 (0.78–0.89)	<0.001	1.00	0.40 (0.25–0.63)	<0.001	0.37 (0.23–0.59)	<0.001
Model 8	0.86 (0.80–0.92)	<0.001	1.00	0.45 (0.29–0.70)	<0.001	0.44 (0.28–0.69)	<0.001

DM = Diabetes mellitus, MVPA = moderate-to-vigorous physical activity. Model 1: adjusted for age and sex. Model 2: model 1 + adjustment for kidney function (eGFR, urinary protein excretion, time between transplantation and baseline, and primary renal disease). Model 3: model 1 + adjustment for transplant characteristics (acute rejection, pre-emptive transplantation, donor type). Model 4: model 1 + adjustment for lifestyle factors (smoking, alcohol consumption, daily caloric intake). Model 5: model 1 + adjustment for immunosuppressive medication (calcineurin inhibitors, prednisolonee). Model 6: model 1 + adjustment for lipids and blood pressure (systolic blood pressure, use of antihypertensive drugs, triglycerides, HDL-C). Model 7: model 1 + adjustment for BMI and waist circumference. Model 8: model 1 + adjustment for 24-h creatinine excretion. Model 9: model 1 + adjustment for fasting plasma glucose and HbA1c. Model 10: model 1 + adjustment for Mediterranean diet score.

**Table 3 jcm-09-00415-t003:** Additional analysis on the associations of MVPA with long-term health outcomes in RTRs who are working (*n* = 322).

Physical Activity	MVPA (cont.)		No-MVPA		MVPA > 0		
	HR^ (95% CI)	*p*-value	N *	Reference	N *	HR^^ (95% CI)	*p*-Value
Post-transplant DM							
Non-occupational PA	0.87 (0.74–1.03)	0.113	10	1.00	10	0.46 (0.18–1.13)	0.076
Total PA	0.91 (0.78–1.06)	0.212	8	1.00	12	0.48 (0.20–1.20)	0.056
Cardiovascular mortality							
Non-occupational PA	0.63 (0.48–0.83)	0.001	12	1.00	3	0.11 (0.11–0.42)	0.001
Total PA	0.75 (0.63–0.91)	0.003	9	1.00	6	0.23 (0.10–0.58)	0.051
All-cause mortality							
Non-occupational PA	0.76 (0.66–0.87)	<0.01	25	1.00	11	0.21 (0.14–0.51)	<0.01
Total PA	0.82 (0.74–0.92)	0.001	19	1.00	17	0.30 (0.18–0.58)	<0.01

Total PA was the sum of non-occupational and occupational MVPA in min/week. DM = diabetes mellitus, MVPA = moderate-to-vigorous physical activity, HR=hazard ratio, CVD = cardiovascular disease, N * = number of events. ^ Analyses were adjusted for age, gender, and kidney function parameters. ^^ Analyses were adjusted for age and gender (kidney function parameters excluded in this analysis due to fulfill the rule of thumb).

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
