# Peer review of "Physical Activity and the Development of Post-Transplant Diabetes Mellitus, and Cardiovascular- and All-Cause Mortality in Renal Transplant Recipients"

_jcm, 2020, doi:10.3390/jcm9020415_

Round 1

Reviewer 1 Report

t is a well done descriptive study but it would have been useful to know the Cardiovascular therapy of the Patients.
In the absence of functional evaluations on cardiovascular efficiency, a 6 min walking test could be useful. Why hasn't it been done?

Author Response

Comment #1: It is a well done descriptive study but it would have been useful to know the Cardiovascular therapy of the Patients.

Response to comment #1: Thank you for your thoughtful and thorough review of our manuscript. We thank the reviewer for this suggestion. In addition to the already presented data on anti-hypertensive medication and statins, we now extend Table 1 with inclusion of a specific subheading on cardiovascular medication use in the revised version of the manuscript (page 5, line 204):

Table 1. Additions into the Table 1 in the revised version of the manuscript.

Variable

Total

(n=650)

No-MVPA

(n=246)

MVPA-1

(n=201)

MVPA-2

(n=203)

P value

Cardiovascular medication use

A2 antagonist (%, n)

14.8 (96)

15.4 (38)

12.9 (26)

15.8 (32)

0.68

ACE inhibitor (%, n)

32.2 (209)

32.1 (79)

30.3 (61)

34.0 (69)

0.74

RAAS blockers (%, n)

47.8 (311)

49.2 (121)

44.8 (90)

49.3 (100)

0.58

Beta-blockers (%, n)

63.2 (411)

63.4 (156)

62.2 (125)

64.0 (130)

0.93

Calcium channel blockers (%, n)

24.5 (159)

26.0 (64)

20.4 (41)

26.6 (54)

0.27

Diuretics (%, n)

40.0 (260)

52.0 (128)

27.4 (50)

37.9 (77)

0.001

Vitamin K antagonist (%, n)

11.4 (74)

13 (32)

10.4 (21)

10.3 (21)

0.60

mTOR inhibitor (%, n)

1.8 (12)

3.3 (8)

1 (2)

1 (2)

0.60

Anti-diabetic drugs (%, n)

14.8 (96)

18.7 (46)

14.4 (29)

10.3 (21)

0.045

A2=angiotensin 2, ACE= Angiotensin-converting-enzyme, RAAS= Renin–angiotensin–aldosterone system, mTOR=mammalian target of rapamycin.

Comment #2: In the absence of functional evaluations on cardiovascular efficiency, a 6 min walking test could be useful. Why hasn't it been done?

Response to comment #2: We agree with this criticism. Unfortunately we did not perform a 6 min walking test because it was not included in the protocol which was approved by the Medical Ethical Committee. Accordingly, we mentioned this in the limitations paragraph of the discussion section of the revised manuscript (pages 13, lines 403-405):

“…A limitation is that we could not have data on functional evaluations, like e.g. a 6 min walking test, which could have provided important information on cardiovascular efficiency....”

Reviewer 2 Report

This is a very well designed and very well written study on the correlation found between physical activity and posttransplant diabetes and mortality of all causes after renal transplantation in a relatively large cohort.

The analysis is very detailed and authors have taken as many confounding factors as possible into account to exclude bias.

This is a very important study.
My only comment is that this remains an observational study and that no causality can be guaranteed with certainty. Authors acknowledge this at the very end.
One important factor impacting diabetes and outcome is the diet. Previous studies in the field of liver transplant have shown that the combination of physical activity and diet was essential in improving physical outcome.  Physical activity without diet may not necessarily affect diabetes and outcome as well. I was wondering whether authors have taken this important parameter (diet) into account. This  should be discussed.

What is now ,needed in the field (and I would suggest the authors to mention that) are large scale interventional studies, testing the effect ab initio immediately post transplant of prescribed additional physical activity on diabetes mellitus and health outcome parameters and delayed/secondary interventional studies when  diabetes mellitus has already developed.

Author Response

Comment #1: My only comment is that this remains an observational study and that no causality can be guaranteed with certainty. Authors acknowledge this at the very end. 

Response to comment #1: Thank the reviewer for the kind words and for the thoughtful review of our manuscript. To accommodate the comment of the reviewer, we moved mentioning of the fact that this concerns an observational study and that no causality can be guaranteed from the end of the limitations paragraph of the discussion section to the beginning of the limitations paragraph of the discussion secton of the revised manuscript (page 12, lines 391-392).

Comment #2:

One important factor impacting diabetes and outcome is the diet. Previous studies in the field of liver transplant have shown that the combination of physical activity and diet was essential in improving physical outcome.  Physical activity without diet may not necessarily affect diabetes and outcome as well. I was wondering whether authors have taken this important parameter (diet) into account. This  should be discussed.

Response to comment #2: We agree that diet may be an important parameter to be taken into account. To accommodate the comment of the reviewer, in the revised version of the manuscript, we adjusted model 1 relating to the association between MVPA and PTDM for diet quality (Model 10). We furthermore investigated whether diet quality might modify the association of MVPA with development of PTDM by additional inclusion of a product-term of the continuous variables of diet quality and MVPA in the concerned model, to assess potential interaction between the two. We found that the association between MVPA and PTDM became slightly stronger when adjusted for Mediterranean diet score, but there was no effect modification by diet quality (P-interaction=0.147)….. Accordingly, following sentences are added in the revised manuscript: In the methods section (page 4, lines 166-167):

“… We also adjusted model 1 relating to the association between MVPA and PTDM for diet quality (Model 10). We furthermore investigated whether diet quality might modify the association of MVPA with development of PTDM by additional inclusion of a product-term of the continuous variables of diet quality and MVPA in the concerned model, to assess potential interaction between the two. Mediterranean diet score was used as diet quality and assessed with a 177-item validated food frequency questionnaire which is described in greater detail elsewhere [33]. ”

In the result section (page 6, lines 210), we now extend the table 2 adding results of model 10:

Table 2. Additions into the table 2 in the revised manuscript

Physical activity

MVPA (cont.)

No-MVPA

(Ref)

MVPA-1

MVPA-2

HR (95% CI)

P

HR (95% CI)

P

HR (95% CI)

P

Post-transplant DM

Model 10

0.87 (0.78-0.97)

0.01

1.00

0.58 (0.32-1.25)

0.11

0.44 (0.21-0.92)

0.03

DM=Diabetes mellitus, MVPA=moderate-to-vigorous physical activity.

Model 10:   model 2 + Mediterranean diet score

In the discussion (page 11, lines 321-326):

“… Furthermore, diet is an important factor in the development of diabetes. A previous analysis by Osté et al. for our study population showed that Mediterranean style diet prevents the development of PTDM [45]. We found that the association between MVPA and PTDM became slightly stronger when adjusted for Mediterranean diet score indicating the importance of diet, but there was no effect modification by diet quality (P-interaction=0.147)…”

Comment #3:

What is now, needed in the field (and I would suggest the authors to mention that) are large scale interventional studies, testing the effect ab initio immediately post transplant of prescribed additional physical activity on diabetes mellitus and health outcome parameters and delayed/secondary interventional studies when  diabetes mellitus has already developed.

Response to comment #1: We thank the reviewer for this suggestion. Accordingly, we mention it in the conclusion part of the revised manuscript (page 13, lines 422-424):

“…Furthermore, large scale interventional studies are needed to test the ab initio effect of physical activity after transplantation on the development of post-transplant diabetes mellitus…”

Reviewer 3 Report

The manuscript of Byamnasukh et al. is a prospective observational study investigating the association between post-transplant non-occupational daily-life moderate-to-vigorous physical activity (MVPA; assessed at a median of 5.7 years post) and the risk of developing diabetes as well as cardiovascular and all-cause mortality. In addition, the authors assessed the benefits of non-occupational MVPA and total daily-life MVPA, including occupational MVPA. Here are a few comments:

Introduction: The authors stated that there are no specific clinical guidelines or recommendations for renal transplant recipients. I suggest nuancing this sentence, as a position statement on exercise for solid organ transplant candidates and recipients has been released in 2019 (Janaudis-Ferreira et al. Transplantation, 2019;103:e220–e238) that now includes some recommendations for both the adult and pediatric solid organ transplant populations. Introduction: Be careful with the use of the term “body composition”, as you did not accurately measure this variable. BMI and waist circumference are convenient clinical indicators, but do not accurately reflect body composition as compared to DXA for instance. Methods: The authors need to clarify the exclusion criteria. My understanding was that they included all recipients who met their inclusion criteria. How about recipients with other solid organ transplants, ambulatory or significant orthopedic problems, contraindications to exercise testing, cognitive decline, etc. ? This should be made clearer in the manuscript. Analysis: How did you select the variables included in the multivariable models? Were these selected based on the results of the univariate analyses using a preset significance cut-off? Please clarify. Analysis: I would also expect BMI and waist circumference to show collinearity, when inserted in the same model. How did you address this? Analysis: I am concerned by the large number of variables included in the models with respect to the small number of outcome events. For instance, in the Cox models on the incidence of post-transplant diabetes, there are respectively 23, 14 and 13 events in the inactive and MVPA-1 and -2 categories whereas the models may include up to 10 variables. The same comment applies to results displayed in Table 3. Too many explanatory variables may lead to model overfitting, which in turn may lead to misleading estimates. Figure 2. Include results of the log-rank test on the Kaplan-Meier curves. Limitation: The single-center nature of the study should be acknowledged as a limitation as it may preclude generalization of results. Typo: Figure 1 Association between non-occupational MVPA and PTDM, renal function decline and all-cause mortality in RTR. I believe renal function decline should be replaced by CV mortality.

Author Response

Comment #1: Introduction: The authors stated that there are no specific clinical guidelines or recommendations for renal transplant recipients. I suggest nuancing this sentence, as a position statement on exercise for solid organ transplant candidates and recipients has been released in 2019 (Janaudis-Ferreira et al. Transplantation, 2019;103:e220–e238) that now includes some recommendations for both the adult and pediatric solid organ transplant populations.

Response to comment #1: We thank the reviewer for this suggestion. Accordingly, we now nuanced the concerned sentence and cite the suggested paper in the introduction section of the revised version of the manuscript (page 2, lines 59-61):

“… A recent position statement on exercise for solid organ transplant recipients released in 2019 recommending that it is a key step toward raising awareness of the importance of exercise training in the patients among transplant professionals…”.

Comment #2: Introduction: Be careful with the use of the term “body composition”, as you did not accurately measure this variable. BMI and waist circumference are convenient clinical indicators, but do not accurately reflect body composition as compared to DXA for instance.

Response to comment #2: To accommodate the comment of the reviewer, we now use the words “obesity measures” instead of “body composition” in the revised version of manuscript (page 2, line 75-76 and Table 1).

Comment #3: Methods: The authors need to clarify the exclusion criteria. My understanding was that they included all recipients who met their inclusion criteria. How about recipients with other solid organ transplants, ambulatory or significant orthopedic problems, contraindications to exercise testing, cognitive decline, etc. ? This should be made clearer in the manuscript.

Response to comment #3: To accommodate the comment of the reviewer, we now more clearly describe the exclusion criteria in the revised version of the manuscript (page 2, lines 83-84):

“… Further exclusions were apparent systemic diseases, such as malignancies or active infections...”

Comment #4: Analysis: How did you select the variables included in the multivariable models? Were these selected based on the results of the univariate analyses using a preset significance cut-off? Please clarify.

Response to comment #4: We selected variables in the multivariate models based on clinical reasons considering which variables known as potential confounders in the relating associations of physical activity with long-term outcomes. We now clarify the selection of variables included in the multivariable models (page 4, lines 144-145):

“… which are clinically known confounders in the relating associations of physical activity with long-term outcomes [14, 33, 6-7, 12-13]...”.

[14]. Zelle DM, Corpeleijn E, Stolk RP, de Greef MHG, Gans ROB, van der Heide JJH, et al.: Low physical activity and risk of cardiovascular and all-cause mortality in renal transplant recipients. Clin J Am Soc Nephrol 2011;6:898–905.

[33]. Osté MCJ, Corpeleijn E, Navis GJ, Keyzer CA, Soedamah-Muthu SS, Van Den Berg E, et al. Mediterranean style diet is associated with low risk of new-onset diabetes after renal transplantation. BMJ Open Diabetes Res Care 2017;5:e000283.

[6]. Calella P, Hernández-Sánchez S, Garofalo C, Ruiz JR, Carrero JJ, Bellizzi V: Exercise training in kidney transplant recipients: a systematic review. J Nephrol 2019;1–13.

[7]. Zelle DM, Klaassen G, Van Adrichem E, Bakker SJL, Corpeleijn E, Navis G: Physical inactivity: A risk factor and target for intervention in renal care. Nat Rev Nephrol 2017; DOI: 10.1038/nrneph.2016.187

[12]. Zelle DM, Kok T, Dontje ML, Danchell EI, Navis G, Van Son WJ, et al.: The role of diet and physical activity in post-transplant weight gain after renal transplantation. Clin Transplant 2013;27:E484–E490.

[13]. Orazio L, Hickman I, Armstrong K, Johnson D, Banks M, Isbel N: Higher Levels of Physical Activity Are Associated With a Lower Risk of Abnormal Glucose Tolerance in Renal Transplant Recipients. J Ren Nutr 2009;19:304–313.

Comment #5: Analysis: I would also expect BMI and waist circumference to show collinearity, when inserted in the same model. How did you address this?

Response to comment #5: We agree with this criticism. In the pre-analyses, we checked the correlation between BMI and waist circumference (r=0.84, p<0.001 for men and r=0.81, p<0.001 for women). Then we did Cox-regression analysis using separate models adjusted for BMI and waist circumference in order to test the confounding effect of total fat and fat distribution separately. However, we found no big differences except the association between MVPA and PTDM (we discussed it in relating paragraph of PTDM in the discussion). Finally, we decided to show a model including both indication of total fat (BMI) and fat distribution (waist circumference). To accommodate the comment of the reviewer, we now add the results of these analyses as supplemental material to the manuscript with addition of mentioning of the fact that these analyses were performed to assess potential relevance of co-linearity (page 4, lines 153-158):

“… With regard to potential collinearity (model 7), we tested the correlation between BMI and waist circumference (r=0.84, p<0.001 for men and r=0.81, p<0.001 for women). Then we performed Cox-regression analyses using separate models adjusted for BMI and waist circumference separately (Supplementary materials, Table S2). We found no differences relating to the confounding effects of total fat and fat distribution. Finally, we included those variables in the same model (model 7)…”.

Table S2. adjustments for BMI and waist circumference

Physical activity

MVPA (cont.)

No-MVPA

(Ref)

MVPA-1

MVPA-2

HR (95% CI)

P

HR (95% CI)

P

HR (95% CI)

P

Post-transplant DM

Model 7

0.88 (0.79-0.99)

0.03

1.00

0.63 (0.31-1.25)

0.19

0.50 (0.25-1.03)

0.06

Model 7A

0.90 (0.80-0.99)

0.043

1.00

0.63 (0.32-1.25)

0.19

0.55 (0.27-1.10)

0.09

Model 7B

0.89 (0.80-0.99)

0.03

1.00

0.64 (0.32-1287)

0.21

0.52 (0.26-1.06)

0.07

Cardiovascular mortality

No. of events

53/589

26

14

13

Model 7

0.85 (0.75-0.96)

0.01

1.00

0.51 (0.23-1.11)

0.09

0.40 (0.17-0.92)

0.03

Model 7A

0.84 (0.74-0.94)

0.001

1.00

0.46 (0.23-0.96)

0.04

0.35 (0.15-0795)

0.01

Model 7B

0.85 (0.75-0.97)

0.01

1.00

0.51 (0.24-1.18)

0.09

0.40 (0.17-0.39)

0.03

All-cause mortality

No. of events

129/650

76

27

26

Model 7

0.84 (0.78-0.89)

<0.001

1.00

0.40 (0.25-0.63)

<0.001

0.37 (0.23-0.59)

<0.001

Model 7A

0.83 (0.78-0.89)

<0.001

1.00

0.39 (0.25-0.61)

<0.001

0.37 (0.24-0.58)

<0.001

Model 7B

0.84 (0.78-0.89)

<0.001

1.00

0.40 (0.25-0.64)

<0.001

0.38 (0.24-0.60)

<0.001

DM=Diabetes mellitus, MVPA=moderate-to-vigorous physical activity.

Model 7:     model 1 + adjustment for BMI and waist circumference

Model 7A: model 1 + adjustment for BMI

Model 7B: model 1 + adjustment for waist circumference

Comment #6: Analysis: I am concerned by the large number of variables included in the models with respect to the small number of outcome events. For instance, in the Cox models on the incidence of post-transplant diabetes, there are respectively 23, 14 and 13 events in the inactive and MVPA-1 and -2 categories whereas the models may include up to 10 variables. The same comment applies to results displayed in Table 3. Too many explanatory variables may lead to model overfitting, which in turn may lead to misleading estimates.

Response to comment #6: We agree with the reviewer that models may contain up to 10 variables and that too many explanatory variables may lead to model overfitting. To accommodate the comment of the reviewer, we changed all analyses subsequent to model 2, for them to become based on model 1 rather than on model 2, so that models may now include up to 6 rather than up to 10 variables. Because analyses for the continuous variables now include a minimum of 23+14+13=50 events for PTDM, the rule of thumb which allows 1 variable per 7-10 events is now fulfilled for all analyses [1-2]. This change in performance of analyses did not materially affect point estimates and 95% confidence intervals of hazards of any of the analyses.

References:

Vittinghoff, Eric, and Charles E. McCulloch. "Relaxing the rule of ten events per variable in logistic and Cox regression." American journal of epidemiology 165.6 (2007): 710-718. Spaderna, Heike, et al. "Dietary habits are related to outcomes in patients with advanced heart failure awaiting heart transplantation." Journal of cardiac failure 19.4 (2013): 240-250.

Accordingly, following sentence is added in the revised manuscript (page 4, lines 166-167):

“… All models (1-10) include up to 6 variables to fulfill the rule of thumb which allows 1 variable per 7-10 events. This is now fulfilled for all analyses [34-35]…”.

Comment #7: Figure 2. Include results of the log-rank test on the Kaplan-Meier curves.

Response to comment #7: We thank the reviewer for this suggestion. Accordingly, in the revised manuscript, we now added the results of log-rank tests to Figure 2.

Comment #8: Limitation: The single-center nature of the study should be acknowledged as a limitation as it may preclude generalization of results.

Response to comment #8: We agree with the reviewer that this is a limitation and accordingly we have added the following sentences to the limitations paragraph of the discussion section of the revised version of the manuscript (page 13, 406-408):

“…Finally, single-center nature of study with mainly consisted of white people. is unclear whether our findings can be extrapolated to other populations. It would be relevant to repeat our study in other patient populations…”

Comment #9: Typo: Figure 1 Association between non-occupational MVPA and PTDM, renal function decline and all-cause mortality in RTR. I believe renal function decline should be replaced by CV mortality.

Response to comment #9: We thank the reviewer for noticing this. Accordingly, in the revised version of the manuscript, we have corrected this.

Thank you for your thoughtful and thorough review of our manuscript.

Reviewer 4 Report

Dear Authors

The paper is very interesting and well written. The study is important and original. All data are presented clearly.

The main limitation of this study is the use of self-reporting of physical activity, but authors discussed.

I would like to congratulate authors. It is very good paper.

Author Response

We thank the reviewer for the kind words appreciate the time taken for the review of our manuscript.

Round 2

Reviewer 3 Report

I would like to thank the authors for their careful responses to my concerns.

My last concern regards the use of the term obesity measures as a replacement for body composition. This choice of term may be confusing as BMI and waist circumference are not indicators exclusively related to obesity. As such, I suggest the authors to replace obesity measures (and body composition as there are still a few occurrences in the text) by anthropometric measures in tables and throughout the text.

Author Response

Response to comment #1: We thank the reviewer for this suggestion. We now use the words “anthropometric measures” instead of “obesity measures” in the revised version of manuscript. We also carefully checked and replaced “body composition” by “anthropometric measures” in the revised manuscript. We appreciate the time taken for the review of our manuscript.